# BENCHMARKING MULTIMODAL VARIATIONAL AUTOENCODERS: CDSPRITES+ DATASET AND TOOLKIT

## ABSTRACT

Multimodal Variational Autoencoders (VAEs) have been the subject of intense research in the past years as they can integrate multiple modalities into a joint representation and can thus serve as a promising tool for both data classification and generation. Several approaches toward multimodal VAE learning have been proposed so far, their comparison and evaluation have however been rather inconsistent. One reason is that the models differ at the implementation level, another problem is that the datasets commonly used in these cases were not initially designed to evaluate multimodal generative models. This paper addresses both mentioned issues. First, we propose a toolkit for systematic multimodal VAE training and comparison. The toolkit currently comprises 4 existing multimodal VAEs and 6 commonly used benchmark datasets along with instructions on how to easily add a new model or a dataset. Second, we present a disentangled bimodal dataset designed to comprehensively evaluate the joint generation and cross-generation capabilities across multiple difficulty levels. We demonstrate the utility of our dataset by comparing the implemented state-of-the-art models.

## 1 INTRODUCTION

Variational Autoencoders (VAEs) (Kingma & Welling, 2014) have become a multipurpose tool applied to various machine perception tasks and robotic applications over the past years (Pu et al., 2016)(Xu et al., 2017)(Nair et al., 2018). For example, some of the recent implementations address areas such as visual question answering (Chen et al., 2019), visual question generation (Uppal et al., 2021) or emotion recognition (Yang & Lee, 2019). Recently, VAEs were also extended for the integration of multiple modalities, enabling mapping different kinds of inputs into a joint latent space. It is then possible to reconstruct one modality from another or to generate semantically matching pairs, provided the model successfully learns the semantic overlap among them.

Several different methods for joint multimodal learning have been proposed so far and new models are still being developed (Wu & Goodman, 2018)(Shi et al., 2019)(Sutter et al., 2021). Two of the widely recognized and compared approaches are the MVAE model (Wu & Goodman, 2018), utilizing the Product of Experts (PoE) model to learn a single joint distribution of the joint posterior, and MMVAE (Shi et al., 2019), using a Mixture of Experts (MoE). Sutter et al. (2021) also recently proposed a combination of these two architectures referred to as Mixture-of-Products-of-Experts (MoPoE-VAE), which approximates the joint posterior on all subsets of the modalities using a generalized multimodal ELBO.

The versatility of these models (i.e. the possibility to classify, reconstruct and jointly generate data using a single model) naturally raises a dispute on how to assess their generative qualities. Wu & Goodman (2018) used test set log-likelihoods to report their results, Shi et al. (2019) proposed four criteria that measure the coherence, synergy and latent factorization of the models using various qualitative and quantitative (usually dataset-dependent) metrics. Evaluation for these criteria was originally performed on several multimodal benchmark datasets such as MNIST (Deng, 2012), CelebA (Liu et al., 2015), MNIST and SVHN combination (Shi et al., 2019) or the Caltech-UCSD Birds (CUB) dataset (Wah et al., 2011). All of the mentioned datasets comprise images paired either with labels (MNIST, CelebA), other images (MNIST-SVHN, PolyMNIST) or text (CelebA, CUB). Since none of these datasets was designed specifically for the evaluation of multimodal integration and semantic coherence of the generated samples, their usage is in certain aspects limited. More

specifically, datasets comprising real-world images/captions (such as CUB or CelebA) are highly noisy, often biased and do not enable automated evaluation. This makes these datasets unsuitable for detailed analysis of the model's individual capabilities such as joint generation, cross-generation or disentanglement of the latent space. On the other hand, simpler datasets like MNIST and SVHN do not offer different levels of complexity, provide only a few categories (10-digit labels) and cannot be used for generalization experiments.

Due to the above-mentioned limitations of the currently used benchmark datasets and also due to a high number of various implementations, objective functions and hyperparameters that are used for the newly developed multimodal VAEs, the conclusions on which models outperform the others substantially differ among the authors (Shi et al., 2019), (Kutuzova et al., 2021), (Daunhawer et al., 2022). Recently, Daunhawer et al. (2022) published a comparative study of multimodal VAEs where they conclude that new benchmarks and a more systematic approach to their evaluation are needed.

In this paper, we propose a unification of the various implementations into a single toolkit which enables training, evaluation and comparison of the state-of-the-art models and also faster prototyping of new methods due to its modularity. The toolkit is written in Python and enables the user to quickly train and test their model on arbitrary data with an automatic hyperparameter grid search. By default, the toolkit includes 6 currently used datasets that have been previously used to compare these models. Moreover, we include our new synthetic dataset that enables a more systematic evaluation called CdSprites+ (*Captioned disentangled Sprites +*) and is built on top of the existing dSprites (disentangled Sprites) dataset used to assess the capabilities of unimodal VAEs (Matthey et al., 2017). While dSprites comprises grayscale images of 3 geometric shapes of various sizes, positions and orientations, CdSprites+ also includes natural language captions as a second modality and 5 different difficulty levels based on the varying number of features (i.e. varying shapes for Level 1 and varying shapes, sizes, colours, image quadrants and backgrounds in Level 5). CdSprites+ enables fast data generation, easy evaluation and also a gradual increase of complexity to better estimate the progress of the tested models. It is described in greater detail in Section 3. For a brief comparison of the benchmark dataset qualities, see also Table 1.

In conclusion, the contributions of this paper are following:

1. We propose a public toolkit which enables systematic development, training and evaluation of the state-of-the-art multimodal VAEs and their comparison on the commonly used multimodal datasets.

2. We provide a synthetic image-text dataset called CdSprites+ designed specifically for the evaluation of the generative capabilities of multimodal VAEs on 5 levels of complexity.

The toolkit and code for the generation of the dataset (as well as a download link for a ready-to-use version of the dataset) are available on GitHub [1].

## 2 RELATED WORK

In this section, we first briefly describe the state-of-the-art multimodal variational autoencoders and how they are evaluated, then we focus on datasets that have been used to demonstrate the models' capabilities.

### 2.1 MULTIMODAL VAES AND EVALUATION

Multimodal VAEs are an extension of the standard Variational Autoencoder (as proposed by Kingma & Welling (2014)) that enables joint integration and reconstruction of two or more modalities. During the past years, a number of approaches toward multimodal integration have been presented (Suzuki et al., 2016), (Wu & Goodman, 2018), (Shi et al., 2019), (Vasco et al., 2020), (Sutter et al., 2021), (Joy et al., 2022). For example, the model proposed by Suzuki et al. (2016) learns the joint multimodal probability through a joint inference network and instantiates an additional inference network for each subset of modalities. A more scalable solution is the MVAE model (Wu & Goodman, 2018), where the joint posterior distribution is approximated using the product of experts (PoE), exploiting

---

[1]GitHub URL (anonymized)

the fact that a product of individual Gaussians is itself a Gaussian. In contrast, the MMVAE approach (Shi et al., 2019) uses a mixture of experts (MoE) to estimate the joint variational posterior based on individual unimodal posteriors. The MoPoE architecture from Sutter et al. (2021) combines the benefits of PoE and MoE approaches by computing the joint posterior for all subsets of modalities. Another recent extension is the DMVAE model, in which the authors enable the encoders to separate the shared and modality-specific features in the latent space for a disentangled multimodal VAE (Lee & Pavlovic, 2021). While there are also other recently proposed multimodal VAEs (Sutter et al. (2020), Daunhawer et al. (2021), Liang et al. (2022), Palumbo et al. (2023)), in this paper, we highlight the 4 abovementioned models that can be considered representative of the individual approaches as summarized by Suzuki & Matsuo (2022). For example, MMVAE does not learn the joint posterior distribution (while the 3 remaining models do), MoPoE is scalable to multiple modalities only under very high computational costs (compared to the other models) and DMVAE is the only model that learns modality-specific (private) latent variables.

The evaluation of the above-mentioned models has also evolved over time. Wu & Goodman (2018) measured the test marginal, joint and conditional log-likelihoods together with the variance of log importance weights. Shi et al. (2019) proposed four criteria for evaluation of the generative capabilities of multimodal VAEs: *coherent joint generation*, *coherent cross-generation*, *latent factorisation* and *synergy*. All criteria are evaluated both qualitatively (through empirical observation of the generated samples) and quantitatively: by adopting pre-trained classifiers for evaluation of the generated content, by training a classifier on the latent vectors to test whether the classes are distinguishable in the latent space, or by calculating the correlation between the jointly and cross-generated samples using the Canonical Correlation Analysis (CCA).

Besides certain dataset-dependent alternatives, the most recent papers use a combination of the above-mentioned metrics for multimodal VAE comparison (Sutter et al., 2021), (Joy et al., 2022) (Daunhawer et al., 2022), (Kutuzova et al., 2021). Despite that, the conclusions on which model performs the best according to these criteria substantially differ. According to a thorough comparative study from Daunhawer et al. (2022), none of the current multimodal VAEs sufficiently fulfils all of the four criteria specified by Shi et al. (2019). Furthermore, the optima of certain training hyperparameters might be different for each model (as was proven e.g. with the regularisation parameter $\beta$ (Daunhawer et al., 2022)), which naturally raises the need for automated and systematic comparison of these models over a large number of hyperparameters, datasets and training schemes. Sutter et al. (2021) released public code which allows comparison of the MoE, PoE, MoPoE approaches - however, the experiments are dataset-dependent and an extension for other data types would thus require writing a substantial amount of new code.

In this paper, we propose a publicly available toolkit for systematic training, evaluation and comparison of the state-of-the-art multimodal VAEs. Special attention is paid to hyperparameter grid search and automatic visualizations of the learning during training. To our knowledge, it is the only model- and dataset-agnostic tool available in this area that would allow fast implementation of new approaches and their testing on arbitrary types of data.

## 2.2 MULTIMODAL DATASETS

At this time, there are several benchmark datasets commonly used for multimodal VAE evaluation. The majority is bimodal, where one or both modalities are images. In some of the datasets, the bimodality is achieved by splitting the data into images as one modality and the class labels as the second - this simplifies the evaluation during inference, yet such dataset does not enable e.g. testing the models for generalization capabilities and the overall number of classes is usually very low. Examples of such datasets are MNIST (Deng, 2012), FashionMNIST (Xiao et al., 2017) or MultiMNIST (Sabour et al., 2017). Another class are image-image datasets such as MNIST and SVHN (Netzer et al., 2011) (as used in Shi et al. (2019)), where the content is semantically identical and the only difference is the form (i.e. handwritten digits vs. house numbers). This can be seen as integrating two identical modalities with different amounts of noise - while such task might be relevant for some applications, it does not evaluate whether the model can integrate multiple modalities with completely different data representations.

An example of a bimodal dataset with an image-text combination is the CelebA dataset containing real-world images of celebrities and textual descriptions of up to 40 binary attributes (such as *male,*

*smiling, beard* etc.) (Liu et al., 2015). While such a dataset is far more complex, the qualitative evaluation of the generated samples is difficult and ambiguous due to the subjectiveness of certain attributes (*attractive, young, wearing lipstick*) combined with the usually blurry output images generated by the models. Another recently used image-text dataset is the Caltech-UCSD Birds (CUB) dataset (Wah et al., 2011) comprising real-world bird images accompanied with manually annotated text descriptions (e.g. *this bird is all black and has a long pointy beak*). However, these images are too complex to be generated by the state-of-the-art models (as proven by Daunhawer et al. (2022)) and the authors thus only use their features and perform the nearest-neighbour lookup to match them with an actual image (Shi et al., 2019). This also makes it impossible to test the models for generalization capabilities (i.e., making an image of a previously unseen bird).

Besides the abovementioned datasets that have already been used for multimodal VAE comparison, there are also other multimodal (mainly image-text) datasets available such as Microsoft COCO (Lin et al., 2014) or Conceptual Captions (Sharma et al., 2018). Similar to CUB, these datasets include real-world images with human-made annotations, which can be used to estimate how the model would perform on real-world data. However, they cannot be used for deeper analysis and comparison of the models as they are very noisy (including typos, wrong grammar, many synonyms, etc.), subjective, biased or they require common sense (e.g., "piercing eyes", "clouds in the sky" or "powdered with sugar"). This makes the automation of the semantic evaluation of the generated outputs difficult. A more suitable example would be the Multimodal3DIdent dataset (Daunhawer et al., 2023), comprising synthetic images with textual descriptions adapted from CLEVR (Johnson et al., 2017). However, this dataset does not have distinct levels of difficulty that would enable to distinguish what the models can learn and what is too challenging.

In conclusion, the currently used benchmark datasets for multimodal VAE evaluation are not optimally suited for benchmarking due to oversimplification of the multimodal scenario (such as the image-label combination or image-image translation where both images show digits), or, in the opposite case, due to overly complex modalities that are challenging to reconstruct and difficult to evaluate. In this paper, we address the lack of suitable benchmark datasets for multimodal VAE evaluation by proposing a custom synthetic dataset called CdSprites+ (*Captioned disentangled Sprites +*). This dataset extends the unimodal dSprites dataset (Matthey et al., 2017) with natural language captions, additional features (such as colours and textures) and 5 different difficulty levels. It is designed for fast data generation, easy evaluation and also for the gradual increase of complexity to better estimate the progress of the tested models. It is described in greater detail in Section 3. For a brief comparison of the benchmark dataset qualities, see also Table 1.

## 3  CdSprites+ Dataset

Figure 1: Examples of our proposed CdSprites+ dataset. The dataset contains RGB images (left columns) and their textual descriptions (right columns). We provide 5 levels of difficulty (*left* to *right*). Level 1 only varies the shape attribute, Level 2 varies shape and size, Level 3 varies also the colour attribute, Level 4 varies the position and Level 5 varies also the background shade. See a more detailed description of the dataset in Section 3.

We propose a synthetic image-text dataset called CdSprites+ (*Captioned disentangled Sprites +*) for a clear and systematic evaluation of the multimodal VAE models. The main highlights of the dataset are its scaled complexity, quick adjustability towards noise or the number of classes and its rigid structure which enables automated qualitative and quantitative evaluation. This dataset extends the unimodal dSprites dataset (Matthey et al., 2017) with natural language captions, additional features

(such as colours and textures) and 5 different difficulty levels. A ready-to-use version of the dataset can be downloaded on the link provided in our repository [1], the users can also modify and generate the data on their own using the code provided in our toolkit.

## 3.1 DATASET STRUCTURE

The dataset comprises images of geometric shapes (64x64x3 pixels) with a defined set of 1 - 5 attributes (designed for a gradual increase of complexity) and their textual descriptions. The full variability covers 3 shape primitives (heart, square, ellipse), 2 sizes (big, small), 5 colours, 4 locations (top/bottom + left/right) and 2 backgrounds (dark/light), creating 240 possible unique feature combinations (see Fig. 1 for examples and Table 5 in the Appendix for the statistics and Section A.1 for further information). To avoid overfitting, the colours of the shapes are textured as well as the changing backgrounds in level 5. Another source of noise (besides textures) is the randomised positions and orientations of the shapes. The default version of the dataset consists of 75k (Level 1) 120k (Level 2), 300k (Level 3), 600k (Level 4) and 960k (Level 5), samples (where 10 % is used for validation).

The textual descriptions comprise 1 - 8 words (based on the selected number of attributes) which have a rigid order within the sentence (i.e. size, colour, shape, position and background colour). To make this modality challenging to learn, we do not represent the text at the word level, but rather at the character level. The text is thus represented as vectors $[x_1, ..., x_N]$ where $N$ is the number of characters in the sentence (the longest sentence has 45 characters) and $x_{1,...,N}$ are the one-hot encodings of length 27 (full English alphabet + space). The total sequence length is thus different for each textual description - this is automatically handled by the toolkit using zero-padding and the creation of masks (see Section 5). A PCA visualization of CdSprites+ Level 5 for both image and text modalities is shown in the Appendix in Fig. 6.

## 3.2 SCALABLE COMPLEXITY

To enable finding the minimal functioning scenario for each model, the CdSprites+ dataset can be generated in multiple levels of difficulty - by the difficulty we mean the number of semantic domains the model has to distinguish (rather than the size/dimensionality of the modalities). Altogether, there are 5 difficulty levels (see Fig. 1):

- **Level 1** - the model only learns the shape names (e.g. *square*)
- **Level 2** - the model learns the shape and its size (e.g. *small square*)
- **Level 3** - the model has to learn the shape, its size and textured colour (*small red square*)
- **Level 4** - the model also has to learn the position (*small red square at top left*)
- **Level 5** - the model learns also the background shade (*small red square at top left on dark*)

We provide a detailed description of how to configure and generate the dataset on our Github repository [1].

## 3.3 AUTOMATED EVALUATION

The rigid structure and synthetic origin of the CdSprites+ dataset enable automatic evaluation of the coherence of the generated samples. The generated text can be tokenized and each word can be looked up in the dataset vocabulary based on its position in the sentence (for example, the first word always has to refer to the size in case of Level 5 of the dataset). The visual attributes of the generated images are estimated using pre-trained image classifiers specified for the given feature and difficulty level. Based on these metrics, it is possible to calculate the joint- and cross-generation accuracies for the given data pair - the image and text samples are considered coherent only if they correspond in all evaluated attributes. The fraction of semantically coherent (correct) generated pairs can thus serve as a percentual accuracy of the model.

We report the joint- and cross-generation accuracies on three levels: *Strict*, *Features* and *Letters* (only applicable for text outputs). The *Strict* metrics measure the percentage of completely correct samples, i.e. there is zero error tolerance (all letters and all features in the image must be correct). The *Features* metrics measure the ratio of correct features per sample (words for the text modality and visual features for the image modality), i.e. accuracy ratio 1.75(0.5)/5 on the level 5 means

Table 1: Comparison of the currently used bimodal benchmark datasets for multimodal VAE evaluation. We compare the overall number of categories (we show the numbers for each level for CdSprites+), the dataset's overall size (we show the size of MNIST and the size of SVHN for MNIST-SVHN and the 5 Levels for CdSprites+), the domains of the two modalities, how the quality of the output can be evaluated (coherency is a more precise estimate compared to the vague canonical correlation analysis (CCA)) and the number of difficulty levels.

| Dataset | Categories | Size | Data Domains | Qualitative Evaluation | Difficulty Levels |
|---------|-----------|------|--------------|------------------------|-------------------|
| MNIST | 10 digits | 60k | image/label | coherency | 1 |
| FashionMNIST | 10 articles | 70k | image/label | coherency | 1 |
| MNIST-SVHN | 10 digits | 60k/600k | image/image | coherency | 1 |
| CelebA | 40 binary | 200k | image/labels | coherency | 1 |
| CUB | 200 bird types | 12k | image/text | CCA | 1 |
| **CdSprites+ (ours)** | 3/6/30/120/240 | 75/120/300/ 600/960k | image/text | coherency | 5 |

that on average $1.75 \pm 0.5$ features out of 5 are recognized correctly for each sample, and *Letters* measures the average percentage of correct letters in the text outputs (see the Appendix Sec. A.2.3 for a detailed description of the metrics).

## 4 BENCHMARKING TOOLKIT

Our proposed toolkit [1] was developed to facilitate and unify the evaluation and comparison of multimodal VAEs. Due to its modular structure, the tool enables adding new models, datasets, encoder and decoder networks or objectives without the need to modify any of the remaining functionalities. It is also possible to train unimodal VAEs to see whether the multimodal integration distorts the quality of the generated samples.

The toolkit is written in Python, the models are defined and trained using the PyTorch Lightning library (Falcon & Team, 2019). For clarity, we directly state in the code which of the functions (e.g. various objectives) were taken from previous implementations so that the user can easily look up the corresponding mathematical expressions.

Currently, the toolkit incorporates the MVAE (Wu & Goodman, 2018), MMVAE (Shi et al., 2019), MoPoE-VAE (Sutter et al., 2021) and DMVAE (Lee & Pavlovic, 2021) models. All of the selected models are trained end-to-end and are able to handle missing modalities on the input, which we consider the basic requirement. You can see an overview of the main differences among the models adopted from Suzuki & Matsuo (2022) in the Appendix Table 16.

The datasets supported by default are MNIST-SVHN, CUB, CelebA, Sprites (a trimodal dataset with animated game characters), FashionMNIST, PolyMNIST and our CdSprites+ dataset. We also provide instructions on how to easily train the models on any new dataset. As for the encoder and decoder neural networks, we offer fine-tuned convolutional networks for image data (for all of the supported datasets) and several Transformer networks aimed at sequential data such as text, image sequences or actions (these can be robotic, human or any other kind of actions). All these components are invariant to the selected model and changing them requires only a change in the config file.

Although we are continuously extending the toolkit with new models and functionalities, the main emphasis is placed on providing a tool that any scientist can easily adjust for their own experiments. The long-term effort behind the project is to make the findings on multimodal VAEs reproducible and replicable. We thus also have tutorials on how to add new models or datasets to the toolkit in our GitHub documentation [2].

### 4.1 EXPERIMENT CONFIGURATION

The training setup and hyperparameters for each experiment can be defined using a YAML config file. Here the user can define any number and combination of modalities (unimodal training is also possible), modality-specific encoders and decoders, desired multimodal integration method,

---

[2]Documentation URL (anonymized)

reconstruction loss term, objective (this includes various subsampling strategies due to their reported impact on the model convergence (Wu & Goodman, 2018), (Daunhawer et al., 2022)) and several training hyperparameters (batch size, latent vector size, learning rate, seed, optimizer and more). For an example of the training config file, please refer to the toolkit repository [1].

## 4.2 EVALUATION METHODS

We provide both dataset-independent and dataset-dependent evaluation metrics. The dataset-independent evaluation methods are the estimation of the test log-likelihood, plots with the KL divergence and reconstruction losses for each modality, and visualizations of the latent space using T-SNE and latent space traversals (reconstructions of latent vectors randomly sampled across each dimension). Examples are shown in the Appendix.

The dataset-dependent methods focus on the evaluation of the 4 criteria specified by Shi et al. (2019): *coherent joint generation*, *coherent cross-generation*, *latent factorisation* and *synergy*. For qualitative evaluation, joint- and cross-generated samples are visualised during and after training. For quantitative analysis of the generated images, we provide the Fréchet Inception Distance (FID) estimation. For our CdSprites+ dataset, we offer an automated qualitative analysis of the joint and cross-modal coherence - the simplicity and rigid structure of the dataset enable binary (correct/incorrect) evaluation for each generated image (with the use of pre-trained classifiers) and also for each letter/word in the generated text. For more details, see Section 3.3.

Table 2: Marginal log-likelihoods (lower is better) for the four models trained on the CelebA dataset using our toolkit. Variance calculated over 3 seeds is shown in brackets.

| Metric | MMVAE | MVAE | MoPoE | DMVAE |
|---|---|---|---|---|
| $logp(x_1)$ | -6239.4 (1.2) | **-6238.2 (1.6)** | -6241.3 (1.8) | -6243.4 (1.3) |
| $logp(x_1, x_2)$ | **-6236.3 (0.9)** | -6238.8 (1.1) | -6242.4 (2.3) | -6239.7 (1.2) |
| $logp(x_1|x_2)$ | -6236.6 (1.6) | -6235.7 (1.3) | **-6235.5 (1.2)** | -6236.8 (1.4) |

## 4.3 DATASETS FOR EVALUATION

The users have the following options to evaluate their models:

- **Using the implemented benchmark datasets.** By default, we provide 6 of the commonly used multimodal benchmark datasets such as MNIST-SVHN, CelebA, CUB etc. Training and testing on these models can be defined in a config.

- **Using the CdSprites+ dataset.** We provide links for downloading all the 5 levels of our dataset. However, it is also possible to generate the data manually. CdSprites+ dataset and its variations can be generated using a script provided in our toolkit - it allows the user to choose the complexity of the data (we provide 5 difficulty levels based on the number of included shape attributes), number of samples per category, specific colours and other features. The dataset generation requires only the standard computer vision libraries and can be generated in a matter of minutes or hours (based on the difficulty level) on a CPU machine. You can read more about the dataset in Section 3.

- **Adding a new dataset.** Due to the modular structure of the toolkit, it is possible to add a new dataset class without disrupting other parts of the code. We provide documentation for how to do it in the GitHub pages linked in our repository [1].

## 5 BENCHMARK STUDY

In this section, we demonstrate the utility of the proposed toolkit and dataset on a set of experiments comparing two selected state-of-the-art multimodal VAEs.

### 5.1 EXPERIMENTS

We perform a set of experiments to compare the four widely discussed multimodal VAE approaches - MVAE (Wu & Goodman, 2018), with the Product-of-Experts multimodal integration, and MMVAE (Shi et al., 2019), presenting the Mixture-of-Experts strategy, MoPoE (Sutter et al., 2021) and DMVAE (Lee & Pavlovic, 2021). Firstly, to verify that the implementations are correct, we replicated some experiments presented in the original papers inside our toolkit. We used the same encoder and

Table 3: Results for the four models trained on MNIST-SVHN using our toolkit. We show the digit classification accuracies (%) of latent variables (*MNIST*) and (*SVHN*), and the probability of digit matching (%) for cross- and joint-generation. The variance of the results calculated over 3 seeds is shown in brackets.

| Version | MNIST | SVHN | MNIST $\rightarrow SVHN$ | SVHN $\rightarrow MNIST$ | Joint |
|---------|-------|------|-----------------|-----------------|-------|
| MMVAE | 85.6 (5.2) | 86.4 (4.6) | **85.7 (5.2)** | **84.5 (4.9)** | 45.3 (3.1) |
| MVAE | **91.5 (4.6)** | **88.2 (3.5)** | 95.4 (2.1) | 83.52 (3.6) | **55.6 (2.6)** |
| MoPoE | 86.6 (3.8) | 83.6 (4.3) | 81.2 (3.6) | 81.8 (2.7) | 43.2 (3.9) |
| DMVAE | 78.27 (4.3) | 69.34 (5.6) | 84.5 (4.7) | 82.2 (3.1) | 44.9 (3.6) |

Table 4: Results for the four models trained on all 5 levels of our CdSprites+ dataset. *Strict* refers to the percentage of completely correct samples (sample pairs in joint generation), *Features* shows the ratio of correct features (i.e., 1.2 (0.1)/3 for Level 3 means that on average $1.2 \pm 0.1$ features out of 3 are recognized correctly for each sample) and *Letters* shows the mean percentage of correctly reconstructed letters (computed sample-wise). Standard deviations over 3 seeds are in brackets. For each model, we chose the most optimal latent space dimensionality *Dims* (for DMVAE a sum of private and shared latents). Please see the Appendix for a detailed explanation of our metrics.

| Level | Model | Dims | Txt→Img Strict [%] | Txt→Img Features [ratio] | Img→Txt Strict [%] | Img→Txt Features [ratio] | Img→Txt Letters [%] | Joint Strict [%] | Joint Features [ratio] |
|-------|-------|------|-----|-----|-----|-----|-----|-----|-----|
| 1 | MMVAE | 16 | 47 (14) | 0.5 (0.1)/1 | 64 (3) | **0.6 (0.0)/1** | **88 (2)** | **17 (10)** | **0.2 (0.1)/1** |
| | MVAE | 16 | **52 (3)** | **0.5 (0.0)/1** | 63 (8) | 0.6 (0.1)/1 | 86 (2) | 5 (9) | 0.1 (0.1)/1 |
| | MoPoE | 16 | 33 (3) | 0.3 (0.0)/1 | 10 (17) | 0.1 (0.2)/1 | 26 (7) | 16 (27) | 0.2 (0.3)/1 |
| | DMVAE | 30 | 33 (4) | 0.3 (0.0)/1 | 4 (5) | 0.0 (0.0)/1 | 25 (2) | 4 (6) | 0.0 (0.1)/1 |
| 2 | MMVAE | 16 | **18 (4)** | 0.8 (0.1)/2 | 41 (20) | 1.4 (0.2)/2 | 85 (4) | **3 (3)** | **0.6 (0.1)/2** |
| | MVAE | 12 | 16 (1) | 0.8 (0.0)/2 | **55 (27)** | **1.5 (0.3)/2** | **91 (6)** | 1 (1) | 0.3 (0.3)/2 |
| | MoPoE | 16 | 10 (3) | 0.8 (0.0)/2 | 8 (7) | 0.7 (0.1)/2 | 40 (4) | 1 (1) | 0.2 (0.1)/2 |
| | DMVAE | 30 | 15 (2) | 0.8 (0.0)/2 | 4 (1) | 0.4 (0.0)/2 | 30 (2) | 0 (0) | 0.2 (0.1)/2 |
| 3 | MMVAE | 16 | 6 (2) | 1.2 (0.2)/3 | 2 (3) | 0.6 (0.2)/3 | 31 (5) | 0 (0) | 0.4 (0.1)/3 |
| | MVAE | 32 | **8 (2)** | **1.3 (0.0)/3** | 59 (4) | **2.5 (0.1)/3** | **93 (1)** | 0 (0) | **0.5 (0.1)/3** |
| | MoPoE | 24 | 7 (4) | 1.3 (0.1)/3 | 0 (0) | 0.7 (0.1)/3 | 32 (0) | 0 (0) | 1.1 (0.1)/3 |
| | DMVAE | 30 | 4 (0) | 1.2 (0.0)/3 | 0 (0) | 0.4 (0.1)/3 | 22 (2) | **1 (1)** | **0.5 (0.1)/3** |
| 4 | MMVAE | 24 | **3 (3)** | **1.7 (0.4)/4** | **1 (2)** | 0.7 (0.4)/4 | **27 (9)** | 0 (0) | 0.5 (0.2)/4 |
| | MVAE | 16 | 0 (0) | 1.3 (0.0)/4 | 0 (0) | 0.2 (0.3)/4 | 16 (5) | 0 (0) | 0.3 (0.6)/4 |
| | MoPoE | 24 | 2 (1) | 1.4 (0.0)/4 | 0 (0) | **0.7 (0.1)/4** | 21 (3) | 0 (0) | 0.1 (0.2)/4 |
| | DMVAE | 30 | 1 (1) | 1.4 (0.0)/4 | 0 (0) | 0.5 (0.1)/4 | 18 (1) | 0 (0) | **0.5 (0.1)/4** |
| 5 | MMVAE | 24 | 0 (0) | 1.8 (0.0)/5 | 0 (0) | 0.1 (0.1)/5 | 13 (2) | 0 (0) | 0.4 (0.1)/5 |
| | MVAE | 16 | 0 (0) | 1.8 (0.0)/5 | 0 (0) | 0.6 (0.0)/5 | **27 (1)** | 0 (0) | 0.2 (0.2)/5 |
| | MoPoE | 24 | 0 (0) | 1.8 (0.0)/5 | 0 (0) | **0.7 (0.0)/5** | 17 (1) | 0 (0) | **1.0 (0.0)/5** |
| | DMVAE | 30 | 0 (0) | 1.8 (0.0)/5 | 0 (0) | 0.6 (0.1)/5 | 18 (2) | 0 (0) | 0.7 (0.1)/5 |

decoder networks and training hyperparameters as in the original implementation and compared the final performance.The results are shown in Appendix Sec. A.3. We then unified the implementations for all models so that they only differ in the modality mixing and trained them on the CelebA and MNIST-SVHN datasets. The results are shown in Tables 2 and 3.

Next, we trained all four models on the CdSprites+ dataset consecutively on all 5 levels of complexity and performed a hyperparameter grid search over the dimensionality of the latent space. You can find the used encoder and decoder architectures (fixed for all models) as well as the specific training details in Appendix Sec. A.2.1. We show the qualitative and quantitative results and discuss them in Section 5.2.

## 5.2 RESULTS

The detailed results for all experiments (including other datasets such as Sprites, MNIST-SVHN etc.) can be found in Appendix Sec. A.2.4. Here we demonstrate the utility of our toolkit and dataset by comparing the MVAE, MMVAE, MoPoE and DMVAE models on the CdSprites+ dataset consecutively on 5 levels of difficulty. For an overview of what attributes each of the levels includes, please see Section 3. In Fig. 2, we show the qualitative results for levels 1, 3 and 5 of the dataset for the

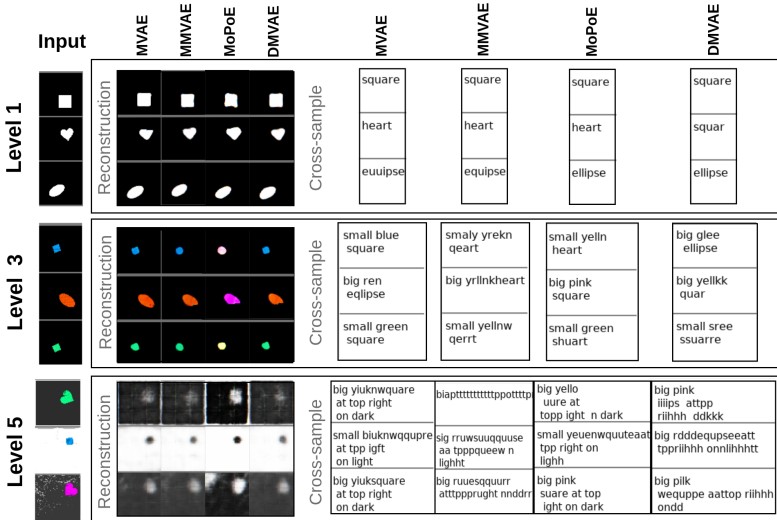

Figure 2: Qualitative results of the MVAE, MMVAE, MoPoE and DMVAE models trained on Level 1, 3 and 5 of our CdSprites+ dataset. We show first the reconstructions of the input image, then the captions obtained by cross-sampling.

four models. We show both the reconstructed and cross-generated (conditioned on the other modality) samples for images and text. Moreover, we report the cross- ($Img \rightarrow Txt$ and $Txt \rightarrow Img$) and joint- ($Joint$) coherency accuracies for all levels in Table 4. The *Strict* metrics show the percentage of completely correct samples, while *Features* and *Letters* show the average proportion of correct words (or visual features for image) or letters per sample.

Based on Table 4, MVAE and MMVAE outperform MoPoE and DMVAE in almost all categories for Levels 1 and 2. While the difference between MVAE and MMVAE is not large in Level 1, MVAE produces far more precise and stable text reconstructions up until Level 3. This would be in accordance with the results by Kutuzova et al. (2021), who showed that the PoE approach outperforms the MoE approach on datasets with a multiplicative ("AND") combination of modalities, which is also the case of our CdSprites+ dataset.

The most prominent trend that can be seen both in Table 4 and Fig. 2 is the gradual performance decline across individual Levels. This is the expected and desirable outcome since only a challenging benchmark allows us to distinguish the difference among the evaluated models. For more details on the results, please see Appendix Sec. A.2.4.

## 6 CONCLUSIONS

In this work, we present a benchmarking toolkit and a CdSprites+ (*Captioned disentangled Sprites+*) dataset for a systematic evaluation and comparison of multimodal variational autoencoders. The tool enables the user to easily configure the experimental setup by specifying the dataset, encoder and decoder architectures, multimodal integration strategy and the desired training hyperparameters all in one config. The framework can be easily extended for new models, datasets, loss functions or the encoder and decoder architectures without the need to restructure the whole environment. In its current form, it includes 4 state-of-the-art models and 6 commonly used datasets.

Furthermore, the proposed synthetic bimodal dataset offers an automated evaluation of the cross-generation and joint-generation capabilities of the multimodal VAE models on 5 different scales of complexity. We also offer several automatic visualization modules that further inform on the latent factorisation of the trained models. We have evaluated the incorporated models on our CdSprites+ dataset and the results are publicly displayed in a "leaderboard" table on the GitHub repository, which will be gradually extended when new models and results are available [1].

The limitation of our CdSprites+ dataset is its synthetic nature and consequent simplicity compared to real-world data. This is a trade-off for the constrained variability allowing a reliable and fully automated evaluation. Please note that our dataset is aimed as a diagnostic tool allowing comparison of performance nuances across various models. We still encourage the researchers to train their models on real-world datasets (which we also include in our toolkit) at the same time to show their full capacity.

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

## A APPENDIX

In Section A.1, we provide additional details for the CdSprites+ dataset. In Section A.2.4, we describe the technical information and detailed results for the experiments presented in the paper and Section A.3 reports the reproduced results from original papers using our toolkit.

## A.1 CDSPRITES+ DATASET STATISTICS

The presented version of the benchmark dataset comprises 5 different levels of difficulty, where each level varies in the number of included features (see Table 5 for their overview). The default size of the dataset is depicted in Table 5 for each level, although the user can easily generate a larger set of the data. The scripts for calculation of the cross- and joint- coherency accuracies use separate batches of testing data provided in the toolkit.

In all levels, the source of noise in the images is their random position and rotation - in levels 1-4, the shapes are located around the whole image with a variance of 25 pixels along both $x$ and $y$ axes. In Level 5, the shapes are shifted in random quadrants where their position also varies with a variance of 8 pixels. The positions are configured so that the whole shapes are always fitting the image. In Levels 2-5 where we vary the size, the proportion of the small objects to the big objects is 1:5.

You can also see a PCA visualization of the CdSprites+ Level 5 dataset in Fig. 6.

Table 5: Statistics of the CdSprites+ benchmark dataset. We show the number of train/validation samples and the number of various shapes, colours, object poses (meaning quadrants which are distinguished in captions) and backgrounds used in each difficulty level. The text captions only describe features that vary (e.g. in level 1, the text descriptions only include the shape name). The colours and backgrounds are all textured when they vary.

| Level | Train Samples | Validation Samples | Shapes | Sizes | Colours | Positions | Backgrounds |
|---|---|---|---|---|---|---|---|
| 1 | 67 500 | 7 500 | 3 | 1 | 1 | 1 | 1 |
| 2 | 108 000 | 12 000 | 3 | 2 | 1 | 1 | 1 |
| 3 | 270 000 | 30 000 | 3 | 2 | 5 | 1 | 1 |
| 4 | 540 000 | 60 000 | 3 | 2 | 5 | 4 | 1 |
| 5 | 864 000 | 96 000 | 3 | 2 | 5 | 4 | 2 |

### A.1.1 USING CHARACTER-WISE EMBEDDINGS

We choose to use character-wise embeddings for CdSprites+ rather than word embeddings. While this choice was made to increase the difficulty of the text modality in our dataset, character embeddings have been recently used also in several other works as this approach brings specific advantages.

Firstly, character-wise embedding does not require a pre-defined vocabulary of possible input words. This can be useful e.g. in incremental learning scenarios where the whole vocabulary is not known prior to training beginning. Secondly, the model can be tested for robustness after training by inputting sentences with misspelt words (e.g., "sqaare" instead of "square") to see if the model can generate correct images. With word-level embedding, this is not possible as replacing entire words will change the feature or create a nonsensical query (e.g., "left square" instead of "blue square").

Please note that we expect the users to use the same encoder and decoder networks (i.e. character transformer networks) for the CdSprites+ benchmark to provide a restricted and fair comparison to other models. Should the users want to use CdSprites+ outside our toolkit for their custom evaluation, they can as well use word-level embeddings as we provide raw strings for the CdSprites+ text modality.

## A.2 BENCHMARK STUDY RESULTS

Here we provide the specific training configuration and hyperparameters used for the experiments on the CdSprites+ dataset as listed in the paper. We also report the detailed results for hyperparameter grid search in terms of the cross- and joint-generation accuracies.

### A.2.1 TRAINING CONFIGURATION

All our experiments were trained with the GeForce GTX 1080 and NVIDIA Tesla V100 GPU cards, the mean computation times for training and inference are shown in Table 6. We used the Adam optimizer, the learning rate of $1e^{-4}$ and all experiments were repeated for 5 seeds (we report standard deviations for the results in the tables). We trained for 150 epochs for Levels 1 and 2 and for 250 epochs in the case of Levels 3-5. In the hyperparameter grid search, we varied the latent dimensionality (16, 24, 32) for all 5 dataset levels and the MVAE, MMVAE and MoPoE models. In the case of DMVAE, the latent dimensionality was different as there are private (modality-dependent) and shared latents. We thus chose different values for the comparison. We used a fixed value of 10 for both private latents and varied the shared latents with values 10, 16 and 24. In Tables 1-5, we

Table 6: Training and inference times for each model trained on our CdSprites+ dataset. The models were trained for 150 epochs on Levels 1-2 and for 250 epochs on Levels 3-5, we thus show these times separately. We show the mean values over all seeds and different latent dimensionalities, the standard deviation is shown as $\pm$.

| Model | Per epoch (s) | Per training (min) (Levels 1-2) | Per training (min) (Levels 3-5) | Inference time (s) |
|-------|---------------|---------------------------------|---------------------------------|--------------------|
| MMVAE | 203 $\pm$2 | 525 $\pm$8 | 860 $\pm$10 | 152 $\pm$6 |
| MVAE | 150 $\pm$20 | 397 $\pm$25 | 645 $\pm$32 | 135 $\pm$19 |
| MoPoE | 127 $\pm$12 | 324 $\pm$9 | 542 $\pm$10 | 126 $\pm$11 |
| DMVAE | 200 $\pm$3 | 500 $\pm$6 | 841 $\pm$7 | 148 $\pm$8 |

show this as the total number of latent dimensions, i.e. 30 (10 shared and $2 \times 10$ private), 36 (16 shared and $2 \times 10$ private) and 46 (24 shared and $2 \times 10$ private).

We used the default training dataset size and validation split as reported in the statistics Table 5. In Tables 7, 8, 9, 10 and 11, we show results for the MVAE, MMVAE, DMVAE and MoPoE models and the compared latent dimensionalities. Standard deviations over 5 seeds are shown in brackets.

### A.2.2 USED ARCHITECTURE

For all evaluated models, we used the standard ELBO loss function with the $\beta$ parameter fixed to 1. For the MVAE [5] model, we used the sub-sampling approach where the model is trained on all subsets of modalities (i.e. images only, text only and images+text). For the image encoder and decoder, we used 4 fully connected layers with ReLU activations. In the case of the text, we used a Transformer network with 8 layers, 2 attention heads, 1024 hidden features and a dropout of 0.1.

### A.2.3 EVALUATION METRICS

After training, we used the script for automated evaluation (provided in our toolkit) to compute the cross- and joint-coherency of the models. For cross-coherency, we generated a 10000-sample test dataset using the dataset generator and used first the images, and then captions as input to the model to reconstruct the missing modality. For joint coherency, we generated 1000 traversal samples over each dimension of the latent space (i.e. 32000 samples for a 32-D latent space) and fed these latent vectors into the models to reconstruct both captions and images.

For both the cross- and joint-coherencies, we report the following metrics: *Strict*, *Feat*, and *Letters* to provide more information on what the models are capable to do. In the first (*Strict*) metrics, we considered the text sample as accurate only if all letters in the description were 100 % accurate, i.e. we did not tolerate any noise. For the image outputs, we considered the images as correct only if all the attributes for the given difficulty level could be detected using our pre-trained classifiers (i.e. correct classification for the shape, colour, size, position or background). For joint coherency, we considered the generated pair as correct only when both the image and captions fulfilled these criteria and were semantically matching.

For the feature-level metrics, we calculated the percentage of correctly reconstructed/generated features (e.g. whole words or image attributes such as shape) and reported the mean percentage of correct features per sample. For the image-caption cross-generation accuracy, we also calculated the average percentage of correct letters per output sample.

In the following section, we report the mean accuracies for both cross- and joint-coherency - these numbers describe the proportion of the correct outputs to all outputs.

### A.2.4 DETAILED RESULTS

In Tables 7, 8, 9, 10 and 11, we show the comparison of the MVAE, MMVAE, DMVAE and MoPoE models on the 5 difficulty levels of the CdSprites+ dataset. Here we varied the latent dimensionality (16-D to 32-D) with the fixed batch size of 32. The values are the mean cross-generation and joint-generation accuracies over 5 seeds with the standard deviations listed in brackets. According to the *Strict* metrics (with zero noise tolerance, see Sec. A.2.3), all models failed in both tasks at Levels 4 and 5. The *Feature* and *Letter* accuracies significantly decrease across levels as the complexity increases. You can see the T-SNE visualizations for the MVAE and MMVAE models trained on Level 4 in Figs. 4 and 5 .

Table 7: Level 1 comparison of accuracies for the four evaluated models trained on our CdSprites+ dataset. *Strict* refers to percentage of completely correct samples (sample pairs in joint generation), *Feats* shows the average percentage of correct features (Level 1 has only 1 feature and *Feats* and *Strict* are thus the same) and *Letters* shows the mean percentage of correctly reconstructed letters.(*Dim*) is the latent space dimensionality.

| Model (Dim) | Txt→Img Strict % | Txt→Img Feats % | Img→Txt Strict % | Img→Txt Feats % | Img→Txt Letters % | Joint Strict % | Joint Feats % |
|---|---|---|---|---|---|---|---|
| MMVAE (16-D) | 47 (14) | N/A | **64 (3)** | N/A | **88 (2)** | **17 (10)** | N/A |
| MVAE (16-D) | **52 (3)** | N/A | 63 (8) | N/A | 86 (2) | 5 (9) | N/A |
| DMVAE (30-D) | 33 (4) | N/A | 4 (5) | N/A | 25 (2) | 4 (6) | N/A |
| MoPoE (16-D) | 33 (3) | N/A | 10 (17) | N/A | 26 (7) | 16 (27) | N/A |
| MMVAE (24-D) | 55 (15) | N/A | 42 (3) | N/A | 31 (12) | 0 (0) | N/A |
| MVAE (24-D) | **55 (4)** | N/A | **61 (3)** | N/A | **82 (1)** | 3 (2) | N/A |
| DMVAE (36-D) | 36 (1) | N/A | 3 (3) | N/A | 21 (2) | **9 (13)** | N/A |
| MoPoE (24-D) | 35 (3) | N/A | 4 (2) | N/A | 24 (6) | 1 (1) | N/A |
| MMVAE (32-D) | 48 (3) | N/A | 36 (2) | N/A | 26 (2) | 0 (0) | N/A |
| MVAE (32-D) | **53 (5)** | N/A | **60 (2)** | N/A | **82 (2)** | **1 (1)** | N/A |
| DMVAE (46-D) | 34 (2) | N/A | 3 (2) | N/A | 20 (9) | 0 (0) | N/A |
| MoPoE (32-D) | 36 (5) | N/A | 2 (1) | N/A | 23 (7) | 0 (0) | N/A |

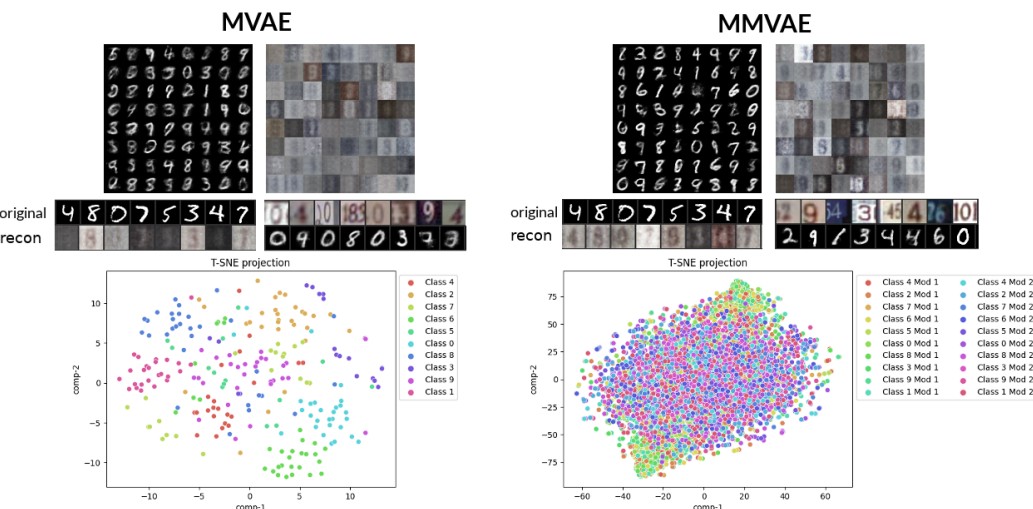

Figure 3: Results for the MVAE and MMVAE models trained on the MNIST-SVHN dataset using our toolkit. For MMVAE, we used the DREG objective as proposed by the authors, MVAE was trained with ELBO. We used the encoder and decoder networks from the original implementations. The top figures are traversals for each modality, below we show cross-generated samples. The bottom figures are T-SNE visualizations of the latent space - please note that for MVAE we show samples from the single joint posterior, while for MMVAE we show samples for both modality-specific distributions.

Table 8: Level 2 comparison of accuracies for the 4 models trained on our CdSprites+ dataset. *Strict* refers to the percentage of completely correct samples (sample pairs in joint generation), *Feats* shows the average percentage of correct features (Level 2 has 2 features) and *Letters* shows the mean percentage of correctly reconstructed letters.(*Dim*) is the latent space dimensionality.

| Model (Dim) | Txt→Img Strict % | Txt→Img Feats % | Img→Txt Strict % | Img→Txt Feats % | Img→Txt Letters % | Joint Strict % | Joint Feats % |
|---|---|---|---|---|---|---|---|
| MMVAE (16-D) | **18 (4)** | 0.8 (0.1)/2 | 41 (20) | 1.4 (0.2)/2 | 85 (4) | **3 (3)** | **0.6 (0.1)/2** |
| MVAE (16-D) | 16 (1) | 0.8 (0.0)/2 | **55 (27)** | **1.5 (0.3)/2** | 91 (6) | 1 (1) | 0.3 (0.3)/2 |
| DMVAE (30-D) | 15 (2) | 0.8 (0.0)/2 | 4 (1) | 0.4 (0.0)/2 | 30 (2) | 0 (0) | 0.2 (0.1)/2 |
| MoPoE (16-D) | 10 (3) | 0.8 (0.0)/2 | 8 (7) | 0.7 (0.1)/2 | 40 (4) | 1 (1) | 0.2 (0.1)/2 |
| MMVAE (24-D) | 17 (5) | 0.4 (0.0)/2 | 16 (0) | 0.4 (0.0)/2 | 40 (2) | 1 (1) | 0.2 (0.0)/2 |
| MVAE (24-D) | 16 (3) | 0.4 (0.0)/2 | **52 (9)** | **0.8 (0.0)/2** | 86 (1) | 5 (6) | 0.3 (0.0)/2 |
| DMVAE (36-D) | **18 (2)** | **0.9 (0.0)/2** | 5 (1) | 0.4 (0.0)/2 | 24 (1) | 0 (0) | 0.2 (0.2)/2 |
| MoPoE (24-D) | 8 (3) | 0.8 (0.0)/2 | 13 (3) | 0.8 (0.1)/2 | 35 (3) | 1 (1) | **0.5 (0.1)/2** |
| MMVAE (32-D) | **17 (1)** | 0.4 (0.0)/2 | 16 (0) | 0.5 (0.0)/2 | 43 (2) | 0 (0) | 0.1 (0.0)/2 |
| MVAE (32-D) | 16 (4) | 0.8 (0.1)/2 | **40 (13)** | **1.8 (0.1)/2** | 87 (1) | 11 (9) | **0.8 (0.0)/2** |
| DMVAE (46-D) | **17 (1)** | **0.8 (0.0)/2** | 3 (1) | 0.4 (0.1)/2 | 24 (2) | 0 (0) | 0.1 (0.1)/2 |
| MoPoE (32-D) | 7 (2) | **0.8 (0.0)/2** | 10 (8) | 0.8 (0.1)/2 | 33 (1) | 0 (0) | 0.3 (0.2)/2 |

Table 9: Level 3 comparison of accuracies for the 4 models trained on our CdSprites+ dataset. *Strict* refers to the percentage of completely correct samples (sample pairs in joint generation), *Feats* shows the average percentage of correct features (Level 3 has 3 features) and *Letters* shows the mean percentage of correctly reconstructed letters.(*Dim*) is the latent space dimensionality.

| Model (Dim) | Txt→Img Strict % | Txt→Img Feats % | Img→Txt Strict % | Img→Txt Feats % | Img→Txt Letters % | Joint Strict % | Joint Feats % |
|---|---|---|---|---|---|---|---|
| MMVAE (16-D) | 6 (2) | 1.2 (0.2)/3 | 2 (3) | 0.6 (0.2)/3 | 31 (5) | 0 (0) | 0.4 (0.1)/3 |
| MVAE (16-D) | 6 (0) | 1.3 (0.0)/3 | **22 (12)** | **2.1 (0.1)/3** | 85 (3) | 0 (0) | **0.5 (0.1)/3** |
| DMVAE (30-D) | 4 (0) | 1.2 (0.0)/3 | 0 (0) | 0.4 (0.1)/3 | 22 (2) | 1 (1) | **0.5 (0.1)/3** |
| MoPoE (16-D) | **6 (1)** | **1.6 (0.0)/3** | 0 (0) | 0.1 (0.1)/3 | 21 (5) | 0 (0) | 0.0 (0.0)/3 |
| MMVAE (24-D) | 4 (3) | 1.2 (0.3)/3 | 1 (1) | 0.8 (0.2)/3 | 31 (6) | 0 (0) | 0.3 (0.0)/3 |
| MVAE (24-D) | **7 (1)** | **1.3 (0.0)/3** | 45 (3) | **2.4 (0.0)/3** | 91 (1) | 0 (0) | **0.6 (0.0)/3** |
| DMVAE (36-D) | 3 (2) | 1.1 (0.1)/3 | 0 (0) | 0.2 (0.0)/3 | 18 (1) | 0 (0) | 0.1 (0.0)/3 |
| MoPoE (24-D) | 7 (4) | 1.3 (0.1)/3 | 0 (0) | 0.7 (0.1)/3 | 32 (0) | 0 (0) | 1.1 (0.1)/3 |
| MMVAE (32-D) | 5 (3) | 1.1 (0.1)/3 | 1 (1) | 0.6 (0.1)/3 | 28 (2) | 0 (0) | 0.0 (0.0)/3 |
| MVAE (32-D) | **8 (2)** | 1.3 (0.0)/3 | 59 (4) | **2.5 (0.1)/3** | 93 (1) | 0 (0) | **0.5 (0.1)/3** |
| DMVAE (46-D) | 5 (1) | 1.1 (0.0)/3 | 0 (0) | 0.1 (0.1)/3 | 15 (1) | 0 (0) | 0.1 (0.0)/3 |
| MoPoE (32-D) | **8 (2)** | **1.5 (0.1)/3** | 0 (1) | 0.6 (0.1)/3 | 28 (1) | 0 (0) | 0.5 (0.2)/3 |

Table 10: Level 4 comparison of accuracies for the 4 models trained on our CdSprites+ dataset. *Strict* refers to the percentage of completely correct samples (sample pairs in joint generation), *Feats* shows the average percentage of correct features (Level 4 has only 4 features) and *Letters* shows the mean percentage of correctly reconstructed letters.(*Dim*) is the latent space dimensionality.

| Model (Dim) | Txt→Img Strict % | Txt→Img Feats % | Img→Txt Strict % | Img→Txt Feats % | Img→Txt Letters % | Joint Strict % | Joint Feats % |
|---|---|---|---|---|---|---|---|
| MMVAE (16-D) | 2 (0) | **1.6 (0.2)/4** | 0 (0) | 0.4 (0.4)/4 | 15 (3) | 0 (0) | 0.1 (0.1)/4 |
| MVAE (16-D) | 0 (0) | 1.3 (0.0)/4 | 0 (0) | 0.2 (0.3)/4 | 16 (5) | 0 (0) | 0.3 (0.6)/4 |
| DMVAE (30-D) | 1 (1) | 1.4 (0.0)/4 | 0 (0) | **0.5 (0.1)/4** | **18 (1)** | 0 (0) | **0.5 (0.1)/4** |
| MoPoE (16-D) | **3 (1)** | **1.6 (0.2)/4** | 0 (0) | **0.5 (0.1)/4** | 16 (3) | 0 (0) | 0.1 (0.1)/4 |
| MMVAE (24-D) | 3 (3) | **1.7 (0.4)/4** | **1 (2)** | 0.7 (0.4)/4 | **27 (9)** | 0 (0) | **0.5 (0.2)/4** |
| MVAE (24-D) | **4 (1)** | 1.2 (0.1)/4 | 0 (1) | **2.4 (0.0)/4** | 14 (1) | 0 (0) | 0.2 (0.1)/4 |
| DMVAE (36-D) | 0 (1) | 1.3 (0.0)/4 | 0 (0) | 0.2 (0.0)/4 | 14 (1) | 0 (0) | 0.2 (0.0)/4 |
| MoPoE (24-D) | 2 (1) | 1.4 (0.0)/4 | 0 (0) | 0.7 (0.1)/4 | 21 (3) | 0 (0) | 0.1 (0.2)/4 |
| MMVAE (32-D) | 1 (1) | 1.6 (0.0)/4 | 0 (0) | 0.9 (0.0)/4 | 21 (0) | 0 (0) | 0.2 (0.0)/4 |
| MVAE (32-D) | 2 (1) | 1.1 (0.1)/4 | 0 (1) | **1.1 (0.0)/4** | 12 (3) | 0 (0) | **0.4 (0.2)/4** |
| DMVAE (46-D) | 1 (1) | 1.2 (0.0)/4 | 0 (0) | 0.1 (0.0)/4 | 14 (1) | 0 (0) | 0.1 (0.0)/4 |
| MoPoE (32-D) | **4 (0)** | **1.7 (0.1)/4** | 0 (0) | 0.5 (0.3)/4 | **20 (3)** | 0 (0) | 0.2 (0.2)/4 |

Table 11: Level 5 comparison of accuracies for the 4 models trained on our CdSprites+ dataset. *Strict* refers to the percentage of completely correct samples (sample pairs in joint generation), *Feats* shows the average percentage of correct features (Level 5 has 5 features) and *Letters* shows the mean percentage of correctly reconstructed letters. (*Dim*) is the latent space dimensionality.

| Model (Dim) | Txt→Img Strict % | Txt→Img Feats % | Img→Txt Strict % | Img→Txt Feats % | Img→Txt Letters % | Joint Strict % | Joint Feats % |
|---|---|---|---|---|---|---|---|
| MMVAE (16-D) | 0 (0) | 1.8 (0.0)/5 | 0 (0) | 0.4 (0.2)/5 | 16 (0) | 0 (0) | 0.7 (0.4)/5 |
| MVAE (16-D) | 0 (0) | 1.8 (0.0)/5 | 0 (0) | **0.6 (0.0)/5** | **27 (1)** | 0 (0) | 0.2 (0.2)/5 |
| DMVAE (30-D) | 0 (0) | 1.8 (0.0)/5 | 0 (0) | 0.6 (0.1)/5 | 18 (2) | 0 (0) | **0.7 (0.1)/5** |
| MoPoE (16-D) | 0 (0) | 1.8 (0.0)/5 | 0 (0) | 0.3 (0.2)/5 | 15 (1) | 0 (0) | 0.5 (0.7)/5 |
| MMVAE (24-D) | 0 (0) | 1.8 (0.0)/5 | 0 (0) | 0.6 (0.1)/5 | 17 (2) | 0 (0) | 0.5 (0.1)/5 |
| MVAE (24-D) | 0 (0) | 1.8 (0.0)/5 | 0 (0) | 0.6 (0.0)/5 | **25 (3)** | 0 (0) | 0.3 (0.0)/5 |
| DMVAE (36-D) | **1 (0)** | 1.8 (0.0)/5 | 0 (0) | 0.6 (0.1)/5 | 14 (0) | 0 (0) | 0.5 (0.1)/5 |
| MoPoE (24-D) | 0 (0) | 1.8 (0.0)/5 | 0 (0) | **0.7 (0.0)/5** | 17 (1) | 0 (0) | **1.0 (0.0)/5** |
| MMVAE (32-D) | 0 (0) | 1.8 (0.0)/5 | 0 (0) | 0.4 (0.1)/5 | 15 (0) | 0 (0) | 0.5 (0.4)/5 |
| MVAE (46-D) | 0 (0) | 1.8 (0.0)/5 | 0 (0) | 0.6 (0.1)/5 | **24 (2)** | 0 (0) | 0.6 (0.1)/5 |
| DMVAE (32-D) | 0 (0) | 1.8 (0.0)/5 | 0 (0) | 0.4 (0.1)/5 | 14 (1) | 0 (0) | 0.4 (0.1)/5 |
| MoPoE (32-D) | 0 (0) | 1.8 (0.0)/5 | 0 (0) | **0.7 (0.3)/5** | 17 (2) | 0 (0) | **1.1 (0.1)/5** |

## MVAE (Level 4)

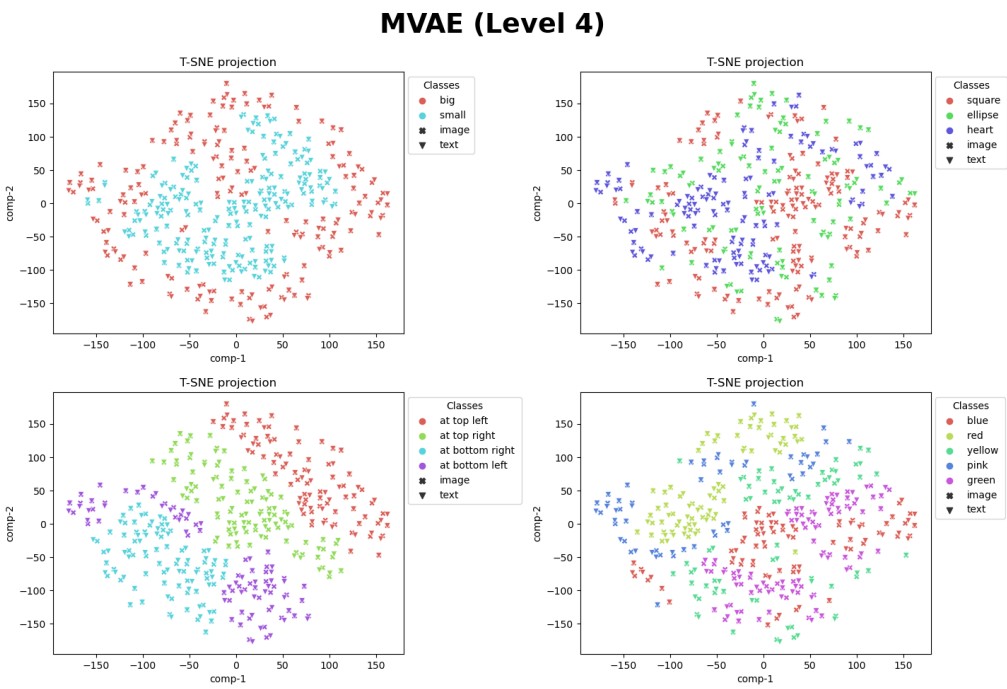

Figure 4: T-SNE visualizations for the MVAE model's (16-D) joint latent space trained on CdSprites+ Level 4. We show the latent space for each of the 4 features (size, shape, position and colour) individually.

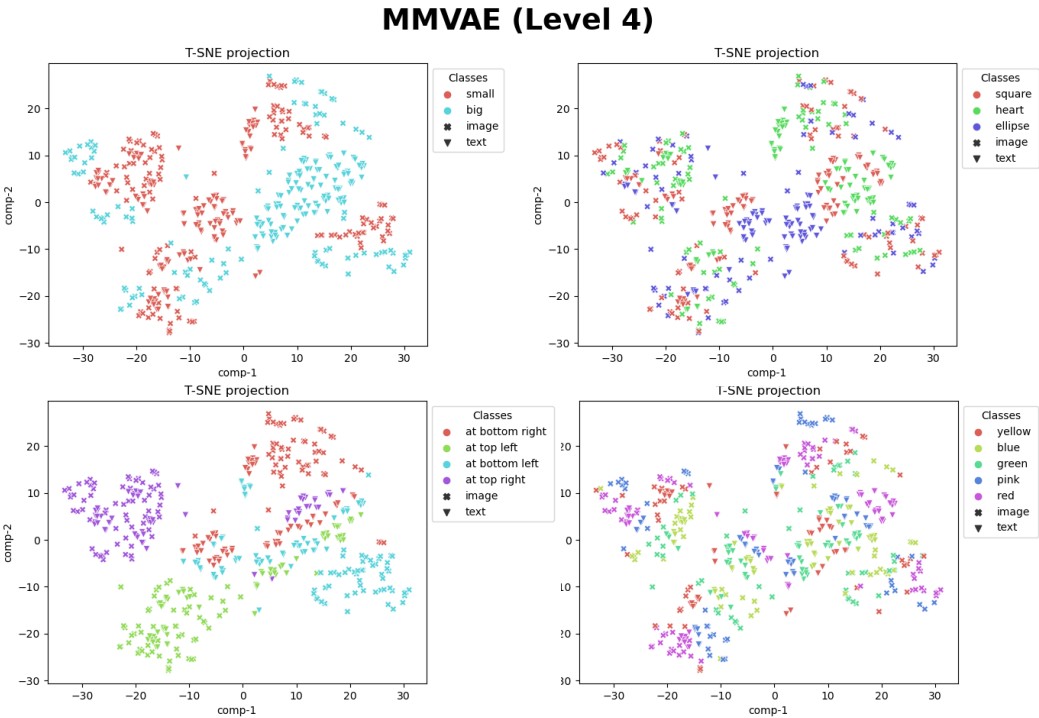

Figure 5: T-SNE visualizations for the MMVAE model's (24-D) unimodal latent spaces trained on CdSprites+ level 4. We show the latent space for each of the 4 features (size, shape, position and colour) individually.

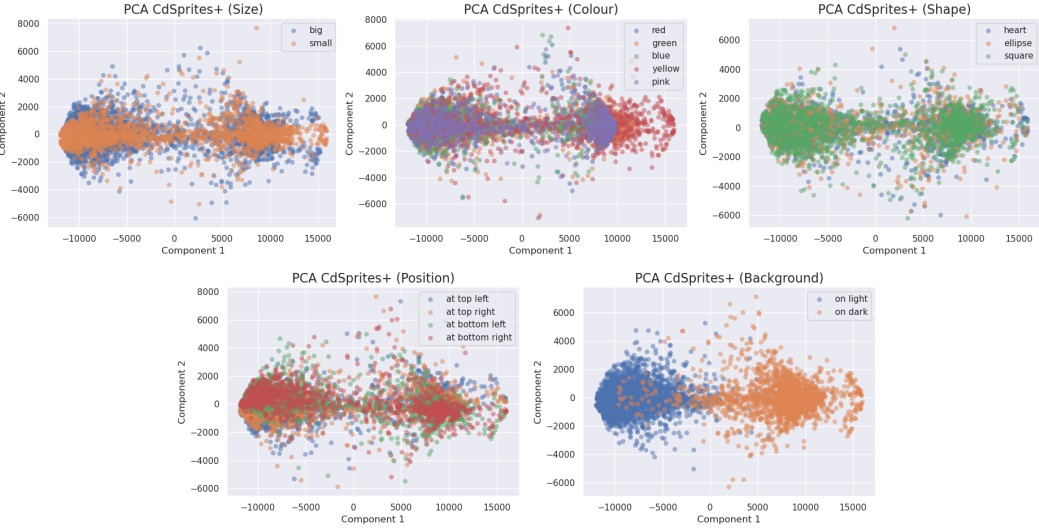

Figure 6: PCA calculated on the images in our CdSprites+ dataset, Level 5. We show a separate figure for each of the 5 features (size, shape, position and colour).

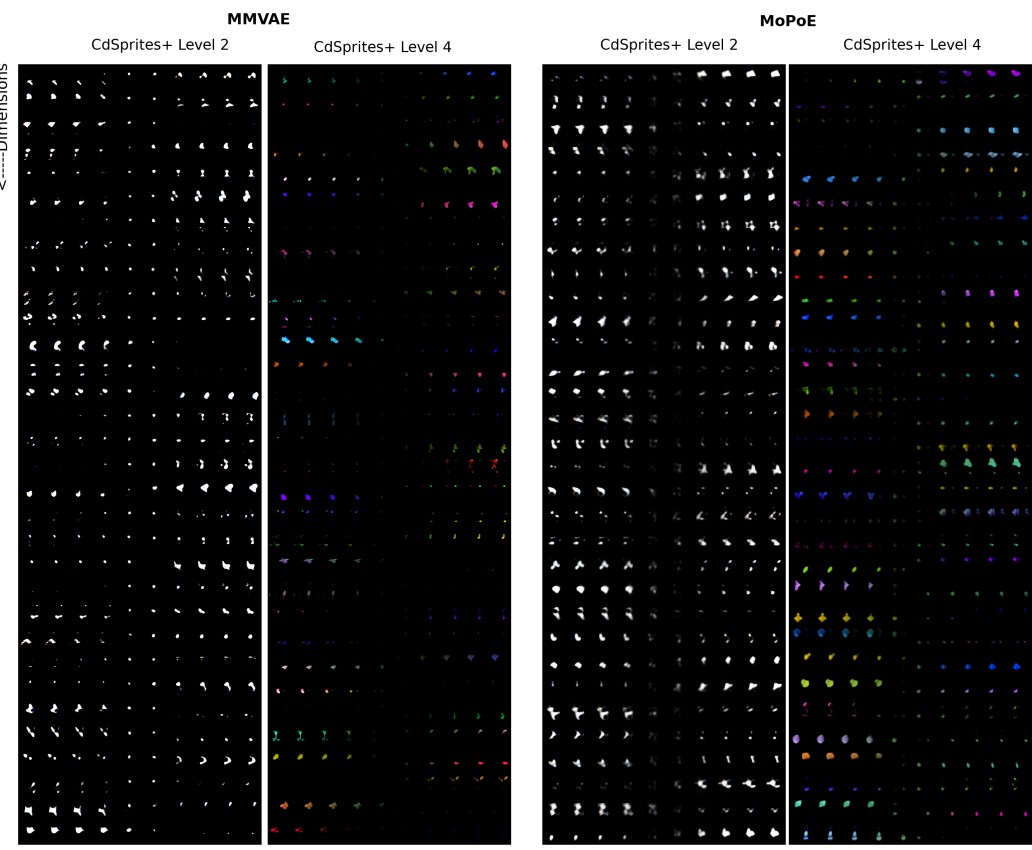

Figure 7: Image traversals for the MMVAE and MoPoE models for the CdSprites+ Levels 2 and 4. Each row is one out of 32 dimensions of the latent space, each column is the single sampled vector from the traversal range (-6,6).

Figure 8: Text traversals for the MVAE and MMVAE models for the CdSprites+ Level 4. Each row is one out of 32 dimensions of the latent space, each column is the single sampled vector from the traversal range (-6,6). Note that we did not set the desired length of the text output, the model thus always generated the maximum number of characters.

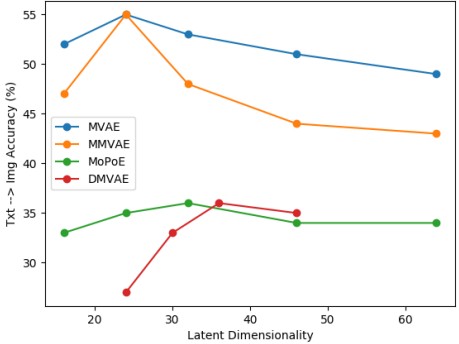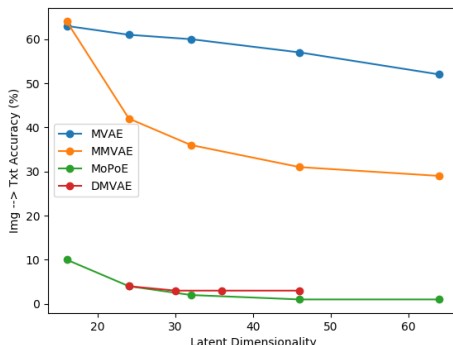

Figure 9: Example of an automated visualization generated by our toolkit. We show accuracies for the Txt → Img (left) and Img → Txt (right) cross-generations for CdSprites+ Level 1 dataset based on the used latent dimensionality. We compare the 4 implemented models. Note that for DMVAE, we trained with different dimensionalities as there are private and shared latent variables.

### A.3 VERIFYING CORRECTNESS OF MODEL IMPLEMENTATION

To verify the correctness of our implementation for each model, we have reproduced selected experiments from the original papers using our toolkit. We provide both the original and our results below.

#### A.3.1 MMVAE

To verify that our implementation of the MMVAE [7] model is correct, we reproduced the experiments using the MNIST-SVHN dataset. We used the same model configuration and parameters as in the original report, i.e. the Mixture-of-Experts mixing with the DREG training objective, latent size 20 and 30 samples drawn from the joint posterior and the likelihood scaling for each modality was adjusted according to the varying dimensionalities. We used the same encoder and decoder architectures as in the original paper. After training, we calculated the joint- and cross-coherencies using the adapted original evaluation script (please see the original paper for the evaluation details [7]). The results are shown in Table 12, the config files for reproducing the experiment are also provided on our GitHub. Please note that the results in Table 12 are different from those depicted in the main paper, Table 3. This is because here we unified the training hyperparameters with the original paper setup. However, we found that setting the likelihood scaling to 1 for both modalities produces more balanced results (in terms of MNIST/SVHN accuracies) and used thus this setup for the comparative study.

#### A.3.2 MVAE

In the original MVAE paper [5], the results are reported in terms of marginal log-likelihoods. We reproduced the FashionMNIST experiment with a 64-D latent space, batch size 100, and likelihood scaling of 10 for the labels and 1 for the images, as reported in the public code. The results can be seen in Table 13.

#### A.3.3 MoPoE

For verification that the MoPoE model is correct, the reproduction was performed on the PolyMNIST dataset. Based on the original implementation, we used the 512-D latent space, Laplace prior distributions, and $\beta = 2.5$. After training, we calculated the cross-coherencies conditioned on 1, 2, 3 or 4 modalities as reported in the paper. The results are shown in Table 14.

#### A.3.4 DMVAE

We reproduced the MNIST-SVHN experiment for the DMVAE model. The reproduced model configuration included shared latent dimensionality $Dim_{shared} = 10$, the private latent dimensionalities

Table 12: Reproduced MNIST-SVHN results for the MMVAE model with our multimodal VAE toolkit. We show the digit classification accuracies (%) of latent variables (*MNIST*) and (*SVHN*), and the probability of digit matching (%) for cross- and joint-generation. For our results, we also show in brackets the variance of the results calculated over 3 seeds.

| Version | MNIST | SVHN | MNIST $\rightarrow SVHN$ | SVHN $\rightarrow MNIST$ | Joint |
|---|---|---|---|---|---|
| Original | 91.3 | 68.0 | 86.4 | 69.1 | 42.1 |
| Reproduced (Ours) | 87.6 (5.2) | 70.4 (4.6) | 82.7 (5.2) | 72.5 (4.9) | 45.3 (3.1) |

Table 13: Reproduced FashionMNIST results for the MVAE model with our multimodal VAE toolkit. We show the estimated marginal log-likelihoods (lower is better). For our results, we also show in brackets the variance of the results calculated over 3 seeds.

| Version | $logp(x_1)$ | $logp(x_1, x_2)$ | $logp(x_1 \vert x_2)$ |
|---|---|---|---|
| Original | -232.535 | -233.007 | -230.695 |
| Reproduced (Ours) | -234.15 (1.52) | -233.89 (2.61) | -232.56 (3.12) |

Table 14: Reproduced PolyMNIST results for the MoPoE model with our multimodal VAE toolkit. We show the Coherence Accuracy (%) of conditionally generated samples (excluding the input modality) (*1 Mod*, *2 Mods*, *3 Mods* and *4 Mods* stand for the number of input modalities) and the joint coherence (*Joint*). For our results, we also show in brackets the variance of the results calculated over 3 seeds.

| Version | 1 Mod | 2 Mods | 3 Mods | 4 Mods | Joint |
|---|---|---|---|---|---|
| Original | 67 | 78 | 80 | 83 | 12 |
| Reproduced (Ours) | 66 (4) | 73 (5) | 81 (3) | 82 (5) | 11 (3) |

Table 15: Reproduced MNIST-SVHN results for the DMVAE model with our multimodal VAE toolkit. We show the probability of digit matching (%) for cross- and joint-generation. For our results, we also show in brackets the variance of the results calculated over 3 seeds.

| Version | MNIST $\rightarrow SVHN$ | SVHN $\rightarrow MNIST$ | Joint |
|---|---|---|---|
| Original | 88.1 | 83.7 | 44.7 |
| Reproduced (Ours) | 84.5 (4.7) | 82.2 (3.1) | 44.9 (3.6) |

Table 16: Comparison of the multimodal VAE models used in our toolkit as described by Suzuki & Matsuo (2022). "Aggregated inference" refers to whether the model learns joint posterior for all modalities, "Modality-specific" means whether the model also learns modality-specific (private) latent variables and "Scalable" refers to whether the computational costs grow exponentially with the number of modalities.

| Model | Aggregated Inference | Modality-specific | Scalable |
|---|---|---|---|
| MVAE (Wu & Goodman, 2018) | ✓ | ✗ | ✓ |
| MMVAE (Shi et al., 2019) | ✗ | ✗ | ✓ |
| MoPoE (Sutter et al., 2021) | ✓ | ✗ | ✗ |
| DMVAE (Lee & Pavlovic, 2021) | ✓ | ✓ | ✓ |

were $Dim_{MNIST} = 1$ and for $Dim_{SVHN} = 4$. The used $\beta$ parameter was 1, and batch size 100. We used the same encoder and decoder networks and an adapted script for calculating the cross- and joint-coherencies. The results are in Table 15.

