# OpenReview forum: "Benchmarking Multimodal Variational Autoencoders: CdSprites+ Dataset and Toolkit"
_ICLR.cc/2024/Conference — ICLR 2024 Conference Desk Rejected Submission_

### Official Review · Reviewer_8MxM · 2023-10-21

**Soundness:** 3 good
**Presentation:** 3 good
**Contribution:** 1 poor
**Rating:** 3
**Confidence:** 5

**Summary:**

The paper proposes a codebase to implement and compare multimodal VAEs in a unified setting. In addition they propose the CdSprites+ dataset, a multimodal variant of dSprites, where images are paired with matching textual descriptions, and a difficulty level can be selected among five different ones.

**Strengths:**

- The paper tackles the relevant problem of evaluation of multimodal VAEs
- The idea of providing a multimodal benchmarking dataset where one can vary the difficulty level is interesting.
- The paper is clear and well-structured.

**Weaknesses:**

In spite of the fact that coming up with better benchmarks for multimodal VAEs is an important problem, I do not think that the relevant aspects of this problem are tackled in this paper. And the proposed benchmark does not bring novelty in the landscape of multimodal datasets used for evaluation. In particular
- PolyMNIST [1] is already a quite successful synthetic benchmark for multimodal VAEs [1,2,3,4]. As a more challenging dataset, Translated PolyMNIST [1], was proposed. Note that the same idea to augment the difficulty of the dataset (making the shared object smaller and shuffle its position in the image) was used in this work. Hence the idea cannot be labelled as novel, and probably it would be necessary to cite [1] in this context.
- The other peculiarity in designing the CdSprites+ dataset is having matching captions as an additional modality. However, the recently proposed Multimodal3DIdent [5][6] is based on a very similar idea, and provides a more realistic setting compared to CdSprites+. In particular, the authors provide the code in a toolbox to generated images such that different versions of the dataset (more and less challenging) can be obtained.
In summary, I think the dataset proposed in this paper does not bring sufficient novel ideas to the community, and the evaluation metrics used (e.g. using pre-trained classifiers for coherence, and FID scores for generative quality) largely build on what is adopted in previous works [7,1,2,3,4].
I think a more interesting side to investigate about benchmarking multimodal VAEs is the need for realistic datasets, where an objective quantitative evaluation of multimodal VAEs is still possible. For instance, general generative coherence is hard to objectively evaluate in real-world scenarios as e.g. CUB Image-Captions [7, 3, 4].

In addition to this, I believe there are other weaknesses in the paper that should be addressed
-Third paragraph in 2.1 seems outdated. A recent work that compares multimodal VAEs comprehensively is [4], where the authors show that their proposed model well-performs when jointly looking at all criteria (generative quality and generative coherence are the main focus, but also latent factorisation and synergy seem to achieved when looking at the results in the Appendix). Note this work is not cited in the paper.
- In spite the aim of the authors is to provide a toolbox to systematically evaluate existing multimodal VAEs, recent relevant work is not implemented [2,4].
- When stating "However, these images are too complex to be generated by the state-of-the-art models (as proven by Daunhawer et al. (2022)) and the authors thus only use their features and perform the nearest-neighbour lookup to match them with an actual image (Shi et al., 2019).". This is not exact as [4] successfully tackle this setting when generating in pixel space.
- Generally, the paper in various parts builds on the assumptions that so far multimodal VAEs have been evaluated inconsistently, while especially more recently certain benchmarks such as PolyMNIST, as well as metrics such as generative coherence with pre-trained classifiers and generative quality via FID scores, are fairly standard and well-established [1,2,3,4].

[1] Thomas M. Sutter, Imant Daunhawer, and Julia E Vogt. Generalized multimodal ELBO. In International Conference on Learning Representations, 2021. [2] HyeongJoo Hwang, Geon-Hyeong Kim, Seunghoon Hong, and Kee-Eung Kim. Multi-view representation learning via total correlation objective. In Advances in Neural Information Processing Systems, 2021. [3] Imant Daunhawer, Thomas M. Sutter, Kieran Chin-Cheong, Emanuele Palumbo, and Julia E Vogt. On the limitations of multimodal VAEs. In International Conference on Learning Representations, 2022. [4] Emanuele Palumbo, Imant Daunhawer, and Julia E Vogt. MMVAE+: Enhancing the generative quality of multimodal VAEs without compromises. In International Conference on Learning Representations, 2023. [5] Imant Daunhawer, Alice Bizeul, Emanuele Palumbo, Alexander Marx, Julia E Vogt. Identifiability Results for Multimodal Contrastive Learning. In International Conference on Learning Representations, 2023. [6]  Alice Bizeul, Imant Daunhawer, Emanuele Palumbo, Bernhard Schölkopf, Alexander Marx, Julia E Vogt. 3DIdentBox: A Toolbox for Identifiability Benchmarking. CLeaR, 2023. [7] Yuge Shi, N. Siddharth, Brooks Paige, and Philip Torr. Variational mixture-of-experts autoencoders for multi-modal deep generative models. In Advances in Neural Information Processing Systems, 2019.

**Questions:**

As a suggestion, as stated above as well, when tackling the problem of enhancing the current state of multimodal VAEs evaluation, I'd encourage the authors to focus more on realistic datasets to objectively evaluate multimodal VAEs, or in coming up with new quantitative proxies for performance criteria that are challenging to evaluate in realistic settings (e.g. semantic coherence).

---

> ### Author Response · Authors · 2023-11-20
> **Our Comment**
>
> Dear reviewer,
>
> Thank you for your comments that help us improve our work.
>
> - The Multimodal3DIdent dataset indeed looks interesting, we were not aware of it as it was developed in parallel with our work and has not yet been evaluated with multimodal VAEs. We have added it to our Related work section and will add it into our toolkit as it is relevant for comparing. The difference from the CdSprites+ dataset and Multimodal3DIdent is mainly that CdSprites+ distinguishes 5 different levels of complexity, that enables quick evaluation of how many distinct features can the model learn at the same time, we thus believe it can be a complimentary benchmark.
> - As for the models included in our toolkit - we have selected 4 models that are representative of the various approaches towards multimodal mixing. We have now added the missing models along with an explanation on how we proceeded with the model selection in Section 2.1 (see also Table 16 in the Appendix).
> - However, please note that the contributions of our paper are not only the dataset, but also the overall toolkit, that enables comparative evaluation of individual models and their development under unified framework. As you are most probably aware, individual multimodal VAE models differ in their implementations and comparison performed with these models are typically done with different loss functions, different hyperparameter settings, etc.. The proposed toolkit enables detailed comparison of the models (currently 4 implemented and easy to add more) with the same settings on currently 7 datasets, so that you can compare the qualitative and quantitative capabilities of these models under different circumstances….The proposed evaluation metrics as well as the option to see traversals (or the newly added comparison of different hyperparameter settings) should make the research in the area of generative models, especially multimodal VAEs, easier and more organized.
> - Furthermore, we believe that our experiments on different multimodal models already show insigthful results, demonstrating the usefulness of the proposed toolkit and dataset. Shortcomings of individual models under various conditions can be easily seen.
> As such, we believe that the proposed toolkit and dataset have enough novelty to be published and used by the community for the future research.
>
> We hope we have addressed your concerns, please feel free to ask if you need any further clarifications. If not, we hope that our answers will help you adjust/conclude on the final score.

---

### Official Review · Reviewer_naTF · 2023-10-27

**Soundness:** 3 good
**Presentation:** 2 fair
**Contribution:** 2 fair
**Rating:** 5
**Confidence:** 4

**Summary:**

The paper proposes a novel benchmark and dataset for evaluating multimodal VAEs. The evaluation consists of two parts: a dataset-dependent part and a dataset-independent part.
In the dataset-independent part, the reconstruction loss (negative log-likelihoods), KL-divergence, and 2d-visualization of latent space embeddings are reported. In the dataset-dependent part, the evaluation focuses on label-based metrics, e.g. latent representation classification accuracy and the coherence of generated samples. 4 different multimodal VAE methods are evaluated.
The proposed dataset is an adapted bimodal version of the well-known dSprites dataset. It offers 5 different levels of increasing complexity of both image and text data.

**Strengths:**

- given the promise of multimodal learning and the ubiquity of multimodal data, establishing benchmarks and benchmarking datasets for multimodal VAEs is important and necessary
- introducing a novel dataset where there are multiple levels of complexity is important for scaling up multimodal VAEs to more difficult tasks and datasets

**Weaknesses:**

- dataset: although I like the dataset with its different complexity levels, I am not sure whether a bimodal dataset is able to fully capture the difficulties of multimodal data. In Sutter et al., 2021, the differences in performance between MVAE, MMVAE, and MoPoE depending on the size of the input subset have been shown. In addition, more than two modalities allow for more complicated relationships between modalities than just information is either shared between all modalities or modality-specific.
- Limited related work: for a benchmark paper, I found the related work section to be a bit short and only a subsection of multimodal VAEs to be discussed. First, there are two (rather) recent review papers on multimodal learning (Liang et al., "Foundations and recent trends in multimodal machine learning: Principles, challenges, and open questions", 2022 [link to review paper](https://arxiv.org/abs/2209.03430), Suzuki and Matsuo, "A survey of multimodal deep generative models", 2022 [link to review paper](https://www.tandfonline.com/doi/full/10.1080/01691864.2022.2035253).
More specifically, there are more works on multimodal VAEs that are similar to the ones discussed:
    - Sutter et al., "Multimodal Generative Learning utilizing the Jensen-Shannon divergence", 2020
    - Daunhawer et al., "Self-supervised Disentanglement of Modality-Specific and Shared Factors Improves Multimodal Generative Models", 2020
    - Palumbo et al., "MMVAE+: Enhancing the Generative Quality of Multimodal VAEs without Compromises", 2023
    - I do not expect that all these methods and models have to be included. However, I would expect them to be listed in the related work section. In addition, it would be good to include the selection criteria for the models/methods implemented in the benchmark.
- for me, to understand the usefulness and especially usability of the proposed benchmark, a diagram of the code structure or some other way to visualize the implemented coding structure would be very helpful. There is no evidence or explanation that tells me what makes it easy to include a novel dataset or method in the benchmark. Anything that supports this would be helpful in my opinion.

**Questions:**

- I wonder whether single best performances for different metrics are the best way to evaluate these models. I would like to have your opinion about reporting metrics for a range of hyperparameters. E.g. the latent representation performance for multiple beta values (e.g. beta 0 [0.1, 0, 1.0, ...])? Or for the latent space dimension. Wouldn't this provide a more thorough picture of a method's performance?
- does it make sense to quantify the coherence of images based on the ratio of correct features? Is this metric able to grasp the semantic coherence?
- image and text are some of the most prevalent modalities. Given the multitude of already existing image-text datasets, as mentioned in the proposed paper, would it not be interesting for a multimodal benchmark to include different modalities and more than just two? I would like to hear your opinion on that.

---

> ### Author Response · Authors · 2023-11-20
> **Questions addressed**
>
> Dear reviewer,
>
> Thank you for your detailed feedback. We comment on each issue below:
>
> 1) **Dataset is only bimodal** - we agree that it would be interesting to also have a benchmark with 3 or more modalities and we plan to develop such dataset in our future work. However, we believe that the detailed analysis performed on the proposed bimodal CdSprites+ dataset already shows significant shortcomings of the current multimodal VAEs and is thus informative as such. It is also possible to easily implement other existing multimodal datasets or to use the trimodal Sprites (with animated characters) which is already included in our benchmark.
> 2) **Limited related work** - unfortunately, the limited number of pages does not allow us to mention all existing multimodal VAEs with at least some minimal description. However, we added more citations for the SOTA models, linked the mentioned surveys and added an explanation of how we selected the models that are currently implemented (please see the updated revision and Table 16 in the Appendix).
> 3) **Diagram of the code** - the diagram of the code can be seen in our GitHub repository: https://github.com/gabinsane/multimodal-vae-comparison#extending-for-own-models-and-networks, the information on what needs to be done to add a new model or benchmark can be found in our Documentation tutorials: https://gabinsane.github.io/multimodal-vae-comparison/docs/html/tutorials/addmodel.html, https://gabinsane.github.io/multimodal-vae-comparison/docs/html/tutorials/adddataset.html
> 4) **Reporting metrics for a range of hyperparameters** - thank you for this useful recommendation. While you can find the extended results for individual latent dimensions in the Appendix (Tables 1-11),  we have now also incorporated code for automated generation of metrics visualization after training, which should make the evaluation more intuitive. We have added an example Fig. 9 in the Appendix which shows how the observed metrics change with the varying latent dimensionality for the individual models. Since the number of hyperparameters is large (e.g., beta, latent dimensionality, number of epochs, reconstruction loss functions, etc.), exploring all combinations would require thousands of trainings. However, such exploration is not the main aim of this paper, it is rather to provide the basis (toolkit, dataset, and evaluation measures) for such a comparison.
> 5) **Quantifying the coherence of images based on the ratio of correct features** - for both the text and the image cross-generation, we calculate how many features that are present in the input modality are also present in the modality that came on the output (for joint generation, we look at the two output modalities). The correct feature ratio thus indeed represents the percentage of semantically coherent features between the two modalities.
>
> We hope we have answered your questions, please feel free to ask if you need any further clarifications. If not, we hope that our answers will help you adjust/conclude on the final score.

---

### Official Review · Reviewer_2Pvg · 2023-10-31

**Soundness:** 3 good
**Presentation:** 3 good
**Contribution:** 3 good
**Rating:** 6
**Confidence:** 3

**Summary:**

The author's present a toolkit for benchmarking multimodal VAEs, a topic where exhaustive and objective model comparison is hard. The tools enables adding new datasets and multimodal VAEs in a modular way

**Strengths:**

As someone that has worked in the topic, I was aware of the lack of libraries that allow for easy comparison of multimodal VAEs. This paper brings tools to solve this.

**Weaknesses:**

A single dataset is included, which limits impact.

How Image quality generation is measured? Could this be adapted to other multimodal generative models?

**Questions:**

See above comments.

---

> ### Author Response · Authors · 2023-11-20
>
> Dear reviewer,
>
> Thank you for your feedback. We answer your concerns and questions below:
>
> **A single dataset is included** - please note that although we propose a single dataset as a systematic benchmark, we also include 6 other standard datasets (as mentioned in Sec. 4.3.) and show results for most of them in the Appendix.
>
> **How is image quality measured** - we evaluate the individual semantic features of the images in CdSprites+ (such as object shape, colour, etc.) using pre-trained image classifiers (see description in Sec. 3.3). For a more general, quantitative analysis of the generated images, we provide the Fréchet Inception Distance (FID) estimation, which is dataset-independent. It is possible to train any other multimodal generative model on our dataset (see provided tutorials in our documentation https://gabinsane.github.io/multimodal-vae-comparison/docs/html/tutorials/adddataset.html ) and get the systematic evaluation as described in the paper (Sec. 3.3).
>
> We hope we have answered your questions, please feel free to ask if you need any further clarifications. If not, we hope that our answers will help you adjust/conclude on the final score.

---

### Official Review · Reviewer_EZ4B · 2023-11-05

**Soundness:** 3 good
**Presentation:** 2 fair
**Contribution:** 2 fair
**Rating:** 5
**Confidence:** 3

**Summary:**

This paper proposes a bimodal benchmark dataset for multimodal VAE study, called Captioned disentangled Sprites (CdSprites). The original Sprites dataset comprises images of objects with different shapes, positions, colors, etc. This proposed new dataset adds captions to the images as the second modality. The tested multimodal VAEs are expected to model both the text and image modality, and are expected to be capable of reconstructing the image and text. Authors chose four multimodal VAEs, MMVAE, MVAE, MoPoE, DMVAE to validate their respective performance on this new dataset, in both prediction and reconstruction power. The new dataset has 5 different levels of difficulty, and an automated visualization tool is provided to analyze models.

**Strengths:**

1. This paper provides a new dataset which contains both image and text modalities, as well as 5 difficulty levels, facilitating the multimodal learning field.
2. The authors take into account 4 state-of-the-art multimodal VAE for evaluation and compared the new dataset with multiple other commonly used datasets.
3. Multiple metrics including prediction, likelihood and reconstruction are incorporated into evaluation.

**Weaknesses:**

1. In Figure 1, regarding the captions, it's a bit vague to tell if the square or heart is big or small unless the model has a sense of the whole dataset.
2. If you use MMVAE, MVAE, MoPoE, DMVAE as 4 representative models for testing, you should elaborate on how are models designed and whey they are representative among all the multimodal VAEs, as well as their distinctions.
3. In section 2.2, while capturing the modalities of image and single class label may be oversimplified, recent works have explored bimodality of feature and multi-label tags [1], which is more challenging while facilitating the test of the alignment in the latent space.
4. While your title says it's multimodal, essentially this paper investigates the bimodal scenario. If you incorporate more modalities, the dataset would be more interesting.
5. While reconstruction and likelihood could be good indicators, for VAE we are more intrigued by the generation power, especially on the text part. I can see some latent traversal results on the image recon from the appendix, but would there be any similar studies on the text side?


[1] Bai, J., et al. Gaussian mixture variational autoencoder with contrastive learning for multi-label classification. In ICML 2022.

**Questions:**

1. To some extent, your level 1-5 is like different granularities regarding the disentangled representation learning. Could you further elaborate on the connections and distinctions between multi-modalities and disentangled representation learning?
2. I can see the cross-sampling qualitative results are less satisfactory. Have you tried to use better text and image encoder before feeding into VAE?
3. It seems that your dataset is somewhat related to some Vision Question Answering (VQA) tasks, which also require capturing both modalities?

---

> ### Author Response · Authors · 2023-11-20
> **Questions addressed**
>
> Dear reviewer,
>
> Thank you for your comments and insight. We discuss the addressed issues below:
>
>
> 1) **The model has to see the whole dataset to know what size is “big” and “small”.** It It is true that the size description is relative and the model has to see the whole train set to learn to distinguish this. The same would apply e.g. to colours or any other continuous feature - the granularity is intentionally decided by the language captions. Since the dataset is randomly shuffled, all categories are introduced from the beginning of the training, so this does not impose any bias. Furthermore, this is an inevitable feature of most image-text datasets, e.g. young/old person in CelebA, long/short legs in the CUB dataset etc.
> 2) **How are the 4 used models representative of the state-of-the-art** - We added an extending paragraph in Sections 2.1 and 4 (both marked blue) to clarify our model choices. We have based our selection on the multimodal VAE survey from [1], who compare the SOTA models based on their key properties: whether they learn a joint latent posterior, whether they can handle missing modalities, whether they learn modality-specific latent variables, whether they are trained end-to-end and whether they are easily scalable toward multiple modalities. We are interested only in models that can handle missing modalities and can be trained end-to-end. We thus chose models that cover the variability across the 3 remaining categories. We have added Table 16 in the Appendix which shows the distinctions among these models and also an additional explanation of the differing features in Section 2.1.
> 3) **There are other multi-label datasets than those mentioned in the Related work** - in Section 2.2, we focused on the datasets that have been used to evaluate multimodal VAEs in the previous works. There are indeed other existing multimodal datasets, which can be also easily adopted (following our provided tutorials) and used inside our toolkit to evaluate the models. However, to our knowledge, none of these datasets allow automated semantic evaluation and provide gradually increasing complexity. Therefore, we propose to use CdSprites+ for the gradually increasing complexity and automatic evaluation of individual features.
> 4) **The proposed dataset is only bimodal** - we agree that it would be interesting to also have a benchmark with 3 or more modalities and we plan to develop such a dataset in our future work. However, we believe that the detailed analysis performed on the proposed bimodal CdSprites+ dataset already shows significant shortcomings of the current multimodal VAEs and is thus informative as it is.
> 5) **Traversals of the text modality** - we have added examples of both text and image traversals for selected models and CdSprites+ levels. Please see the Appendix Figs. 7 and 8. Although the generative quality of the models is important, the traversals can be evaluated only qualitatively. We thus show them as examples of the visualizations produced by our toolkit, but do not rely on them in the comparative study.
> 6) **Connections and distinctions between multi-modalities and disentangled representation learning** - we have created the 5 difficulty levels of CdSprites+ by adding individual features which can and should be each learned in a disentangled manner if the model is capable of doing it. It can thus also inform the user about the maximum number of features the model can learn independently so that their representations are still disentangled within the latent space.
> 7) **Less satisfactory results of cross-sampling** - we have used standard CNN and Transformer networks for the image and text modalities to keep the relatively low computational costs (provided we had to do 300+ trainings). While more recent and larger models might provide higher quality samples, it is not really relevant as we focus on the relative comparison among the models (and we kept the same encoder and decoder networks for all of them).
> 8) **CdSprites+ dataset similar to VQA** - we indeed mention the CLEVR dataset in Section 2.2 which was developed for VQA. However, the long questions used in CLEVR require complex logical reasoning which is currently out of bounds for the state-of-the-art multimodal VAEs. It would be however interesting to add another modality that would query some specific feature of the image (e.g. “What is the size of the shown object?”), and the text modality would only include the selected feature/s. This would however not allow cross-sampling on certain subsets (we then cannot provide the correct caption for the image without the query, etc.).
>
> We hope we have answered your questions, please feel free to ask if you need any further clarifications. If not, we hope that our answers will help you adjust/conclude on the final score.
>
> [1] Suzuki, M., & Matsuo, Y. (2022). A survey of multimodal deep generative models. Advanced Robotics, 36(5-6), 261-278.

---

### Comment · Area_Chair_MLNb · 2023-11-22
**Reject due to the violation of double-blind policy**

Dear authors,

The github link posted in the rebuttal (https://openreview.net/forum?id=3DPTnFokLp&noteId=UMJ526nQw0) has revealed the authors' identity. This is a violation of our policy. The paper is hence rejected.